# A Stable PDLC Film with High Ageing Resistance from an Optimized System Containing Rigid Monomer

**DOI:** 10.3390/molecules28041887

**Published:** 2023-02-16

**Authors:** Hongren Chen, Xiao Wang, Jianjun Xu, Wei Hu, Meina Yu, Lanying Zhang, Yong Jiang, Huai Yang

**Affiliations:** 1School of Materials Science and Engineering, University of Science and Technology Beijing, Beijing 100083, China; 2School of Materials Science and Engineering, Peking University, Beijing 100871, China; 3Institute for Advanced Materials and Technology, University of Science and Technology, Beijing 100083, China

**Keywords:** PDLC, high stability, high temperature, rigid monomer

## Abstract

With the switchability between transparent and light-scattering states, polymer-dispersed liquid crystals (PDLC) are widely used as smart windows, flexible display devices, projectors, and other devices. In outdoor applications, in addition to excellent electro-optical properties, there is also a high demand for film stability. In this work, a PDLC film with high mechanical strength and structural stability is prepared that can maintain stability at 80 °C for 2000 h. By choosing liquid crystals with a wide temperature range, adopting acrylate polymer monomers containing hydroxyl groups, and adjusting the polymer content, the PDLC film can work well from −20 °C to 80 °C. On this basis, the effects of the introduction of rigid monomers on the mechanical properties and electro-optical properties of PDLC films are investigated.

## 1. Introduction

With increasing attention to environmental protection and energy conservation, the research and application of environmentally friendly and smart devices have been a research hotspot [1,2]. For example, smart windows based on electro-, thermo-, gaso-, and photochromogenic materials can change transmittance for blocking intense visible light [3,4,5,6,7]. Benefiting from the large-scale processability, long-term stability, good flexibility, and high mechanical strength, a polymer-dispersed liquid-crystal (PDLC) system has become a mainstream solution [8,9,10,11].

PDLC consists of randomly scattered liquid-crystal (LC) droplets embedded in the polymer matrix [12,13,14,15]. The alignment of LC droplets in the polymer matrix can be changed by the electric field, which can change the transmittance of the PDLC film. Normally, visible light cannot penetrate the PDLC film because of the different refractive indexes between the LC molecules and the polymer matrix. With an electric field, the orientation of LC molecules becomes a regular arrangement so that the PDLC film can be transparent. Meanwhile, the polymer matrix gives PDLC excellent physical and chemical properties, which give the PDLC system potential for the preparation of smart windows [16,17].

For now, the vast majority of research has focused on improving the electro-optical performance of PDLC films, such as driving voltage, contrast ratio (*CR*), and response time. For example, Busbee JD et al. investigated the dispersion behavior of functionalized SiO_2_ nanoparticles (NPs) in a holographic polymer-dispersion liquid-crystal system and found that the high refractive index of SiO_2_ NPs could improve the refractive index difference between the liquid crystal and the polymer matrix, thus improving the CR of PDLC film [18]. Yaroshchuk OV et al. demonstrated that TiO_2_ NPs could modify the contrast and substantially reduce the off-axis haze of PDLC [19]. Guo S et al. proposed a two-step polymerization method to prepare PDLC and a polymer-stabilized liquid-crystal (PSLC) coexistence system, introducing vertically oriented polymer fibers in the polymer network, thus effectively reducing the driving voltage of the film [20].

In practical applications, PDLC films may be used for a long time in harsh environments, such as automobiles and exterior walls of buildings. In the outdoor environment, the long-time high temperature caused by direct sunlight will cause PDLC-aging problems such as unstable performance and shrinkage at the edge position. In this case, in addition to good electro-optical properties, PDLC films need to have a wide range of operating temperatures and good stability. However, there are relatively few studies on the mechanical properties, mechanical strength, and thermodynamic stability of PDLC films.

In this work, the polymer content, liquid crystal, and monomer ratios were optimized to obtain PDLC films with better electro-optical properties and a wide operating temperature range. In addition, different crosslinker and rigid monomer contents were compared to prepare a highly stable electronically controlled dimming film. After testing, the film has a wide working temperature range of −20 °C to 80 °C and also maintains good stability after 2000 h at 80 °C.

## 2. Results and Discussion

### 2.1. Effect of Different LC Contents on Polymer Micromorphology and Mechanical Properties of PDLC Films

The micro-morphology of the polymer network is the main factor affecting the electro-optical properties and mechanical properties of PDLC films, which is mainly affected by the properties of monomers and LCs, the ratio, and polymerization conditions [21]. It has been shown that the LC content is one of the most important factors in determining the polymer network structure, as it highly influences the viscosity of the system on the one hand, and greatly affects the polymerization rate on the other hand [22]. The effect of different LC contents on the polymer micromorphology and mechanical properties of PDLC films were studied in Group A. SEM photographs of the polymer network microstructure are shown in Figure 1. With the decrease in LC content in Samples A1–A5 and the gradual increase in polymerizable monomer content, the polymer network content in the films increased significantly. On the one hand, the polymerization rate slows down when the LC concentration is high, which allows the mixture to remain in the liquid state for a longer time, thus allowing the growth and aggregation of small LC droplets to form larger droplets. On the other hand, the rate of photopolymerization is also high when the monomer content is high, so LC droplet aggregation is low and the polymer network around small LC droplets is more dense [23]. In general, a smaller mesh size means a denser polymer network, which gives the PDLC film better mechanical properties.

Figure 2 illustrates the shear strengths of Samples A1–A5, respectively. In Figure 2a, the film with an effective area of 2.0 cm × 2.0 cm is subjected to an upward and downward stretching force. As shown in Figure 2b, With the gradual increase in polymerizable monomer content in Samples A1–A5, the shear strength of the films also increased significantly. In comparison, when the monomer content reached 50%, the shear strength of Sample A5 reached its maximum value, 182.3 N. Combined with Figure 1, higher polymer matrix content can effectively improve the mechanical strength of the PDLC films. Obviously, in the practical application of PDLC films, the higher the shear strength is, the higher stability the film has and the wider the range of use. Based on this, we choose 50% monomer content as the next experimental condition. 

### 2.2. Effect of Different LCs on the Microscopic Morphology of Polymers and the Electro-Optical Properties of PDLC Films

In addition to the LC content, the different types of liquid crystals also affect the polymer network morphology. We selected three types of liquid crystals for experimental comparison in Group B. The polymer morphology of the prepared samples is shown in Figure 3. Compared to Sample B1, the polymer mesh size in Samples B2 and B3 showed a tendency to become larger. This is because the viscosity of liquid crystals GXP-6011 and GXP-6015 is larger than that of SLC-1717, which reduces the diffusion and aggregation rate of liquid-crystal domains [24]. On the other hand, the micromorphology of the polymer network has a significant effect on the electro-optical properties of PDLC films.

PDLC films can be driven by an electric field for optical-state transition; accordingly, there are some parameters that can be used to characterize the electro-optical properties of the films, such as threshold voltage (*V_th_*) and saturation voltage (*V_sat_*). Typically, they are inversely proportional to the LC droplet radius (*R*), as shown by the following equations [3]:(1)Vth≈dR[K(ω2−1)ε0Δε]12
(2)Vsat≈dR(ω2−1)124πKΔε
where *d*, *K*, *ω*, and *ε_0_* represent the film thickness, elastic constant, aspect ratio, vacuum permittivity, and dielectric anisotropy of the LC, respectively. Generally, a lower driving voltage is conducive to the application of PDLC films. *T_on_* and *T_off_* are defined as the transmittance of the on-state and off-state in PDLC films. CR is the ratio of *T_on_* to *T_off_*. The higher the CR, the more obvious the switch effect of the PDLC film. *τ_on_* and *τ_off_* are defined as the time required for the transmittance to change from 10% to 90% upon turn-on and from 90% to 10% upon turn-off, respectively. The response time, which reflects the sensitivity of PDLC films, can be obtained by adding *τ_on_* and *τ_off_*. Theoretically, they can be calculated according to the following equations:(3)$$\tau_{on}\approx \frac{\gamma }{\Delta \varepsilon V^{2}-\frac{K\left(l^{2}-1\right)}{R^{2}}}$$
(4)$$\tau_{off}\approx \frac{R^{2}\gamma }{K\left(l^{2}-1\right)}$$
where *V* is the applied electric field, and *K*, *l*, *γ*, and Δ*ε* represent the elastic constant, shape anisotropy, rotational viscosity constant and dielectric anisotropy of the liquid crystal, respectively. Accordingly, the increase in the LC droplet radius (*R*) can decrease *τ_on_* and increase *τ_off_* because of the subsequent decrease in anchoring force of the polymer matrix to the liquid-crystal molecules. In summary, the PDLC films suitable for a wide range of applications usually have lower *V_th_* and *V_sat_*, a relatively high *CR*, and a short response time.

The voltage dependence of the transmittance of Samples B1–B3 in Figure 4a shows that with the increase in the applied voltage, the *T_on_* of all samples reaches the saturation level. Figure 4b shows that Sample B2 (GXP-6011) has the lowest driving voltage, according to Equations (1) and (2), which is due to the LC properties and domain size. On the one hand, as the viscosity decreases, the LC molecules are relatively easier to reorient under the electric field. On the other hand, as the polymer mesh-size becomes larger, the specific surface area of the LC domains decreases, and the anchoring force of the LC/polymer interface on the LC molecules decreases [25]. CR is the ratio of *T_on_* to *T_off_*. In Figure 4c, the lower CR of Sample B2 (GXP-6011) compared to the other two samples is due to its relatively high *T_off_*, which is influenced by the LC microdomains and the birefringence of the LCs [26]. Figure 4d shows the response time of all samples. Under the action of an electric field, the LC molecules in all three samples can quickly complete the disorder-to-order orientation transition, and there is no obvious difference. For decay time, Sample B1 (SLC-1717) is relatively the lowest because the LC molecules originally oriented along the electric field revert to the disordered state when the electric field is removed, and the time required for this process is mainly influenced by the LC viscosity and the polymer mesh size [27]. Based on the fact that Sample B2 (GXP-6011) has the lowest drive voltage and the GXP-6011 has a wider phase transition temperature range, we choose the GXP-6011 as the LC of PDLC system and optimize other aspects of tuning in the next step.

### 2.3. Effect of Chain Length of Crosslinker on Polymer Micromorphology and Electro-Optical Properties of PDLC Films

This section explores the effect of crosslinker chain length on the performance of PDLC films. Without changing the components and ratios of other materials in the system, five acrylate monomers PEGDA with different side-chain lengths were compared. The LC droplet size and morphology of the PDLC film are determined during the LC droplet nucleation and the polymer gelation. The LC droplet size is influenced by the rate of polymerization, the relative ratios of the materials, the types of LCs and polymers used, and the physical parameters, such as viscosity, rate of diffusion, and solubility of the LCs in the polymer. As shown in Figure 5, the LC droplet size increased when increasing the chain length of the crosslinking agents. The size of the LC domains is larger because larger chain lengths produce greater spatial site resistance, which leads to a decrease in the polymerization rate. In addition, as the chain length increases, the molar fraction of reactive groups in the system decreases, which slows down the polymerization process and leads to an increase in the pore size of the polymer network. In conclusion, increasing the chain length of the crosslinker leads to an increase in LC droplet size.

Figure 6 shows the electro-optical curves of Samples C1–C5. It could be seen that as the chain length of the crosslinking agent increases, the driving voltage shows a trend of decreasing first and then increasing. This is due to the increasing chain length of the crosslinking agent resulting in a larger polymer matrix mesh, thus lowering the surface anchoring effect between the LCs and the polymer network. On the basis of this, the driving voltage is negatively correlated with the mesh size. We can see in the SEM photos in Figure 5 that the mesh size in Samples C4 and C5 are larger than in Samples C1–C3. However, the driving voltage of Samples C4 and C5 has increased. Based on this, we infer that the increase in driving voltage of Samples C4 and C5 may receive the influence of the chain length of the crosslinker molecules. As the chain length of crosslinker molecules increases, their spatial site resistance increases and their reactivity decreases, which increases the probability that the crosslinker molecules remain in the liquid-crystal microdroplets, thus affecting the electro-optical properties of the liquid-crystal molecules. In Figure 6c, as the chain length of the crosslinking agent increases, the mesh of the polymer changes greatly, causing *CR* to decrease. This is because the light-scattering effect diminishes as the polymer grid size of Samples C1–C5 increases sequentially, leading to an increase in *T_off_*. *T_on_* is mainly influenced by the degree of matching between the *n_p_* of the polymer and the *n_o_* of the LC. Figure 6d shows that the response times of Samples C1–C5 are getting longer, which can be explained by the fact that the LC molecules in the larger mesh take longer to return to the original state when the PDLC films become a scattering state.

### 2.4. Effect of Rigid Polymerizable Monomer Content on the Stability of PDLC Films

Figure 7 shows the photos of the PDLC films with different ratios of Bis-EMA15/PEGDA700. From Samples D1 to D4, the contents of polymerizable monomer Bis-EMA 15 were 2.0%, 4.0%, 6.0%, and 8.0%, respectively (as shown in Table 1). The samples in the pictures were placed in an oven at 80 °C for 2000 h to simulate the environmental conditions of these film samples in actual long-term use. We can see in Sample D1 a large blank area at the edge of the PDLC film, which indicates that the polymer network and LCs in this area have shrunk. Compared with this, the blank area at the boundary of Samples D2–D4 showed a significant decrease. From this, we can infer that the insufficient strength and toughness of the polymer network will cause the PDLC film samples to stretch at the boundaries after a long period of use. As the content of rigid polymerizable monomers (Bis-EMA 15) increases, the strength and toughness of the polymer network increase, which effectively improves the shrinkage problem of PDLC.

Figure 8 shows the SEM photos of the PDLC films with different ratios of Bis-EMA15/PEGDA700. From Samples D1 to D4, the weight ratio of Bis-EMA15/PEGDA700 was 1:4, 2:3, 3:2, and 4:1, respectively (as shown in Table 1). The LC droplet size changed considerably with the change in Bis-EMA15/PEGDA700 content. Since the molecular weight of PEGDA700 is lower than that of Bis-EMA15, the increase in Bis-EMA15 content decreases the molar fraction of reactive groups in the system. Therefore, with the increase in Bis-EMA15 content, the polymerization of monomers becomes slower, which leads to the decrease in crosslink density and the increase in LC droplet size. We can see from Figure 8 that with the decrease in PEGDA700 content and the increase in Bis-EMA15 content, the size of LC microdroplets increased significantly. In general, larger polymer mesh size is not conducive to the mechanical strength and stability of the polymer matrix. In Samples D3 and D4, the polymer mesh size becomes larger, but the stability of these films has improved significantly, which proves that the rigid monomer (Bis-EMA 15) improves the stability more than the mesh-size effect in the PDLC system.

Figure 9 shows the electro-optical curves of Samples D1–D4. In Figure 9a, the variation of the transmittance of PDLC films D1–D4 is a function of the applied voltage. With the increase in applied voltage, all the films achieved the saturation level. We can see from Figure 9b that, with the increase in Bis-EMA15 content, the driving voltage of the film showed a trend of decreasing and then increasing. As mentioned before, a larger polymer mesh size can reduce the driving voltage of the PDLC films because of the reduced anchoring force of the polymer matrix to the liquid-crystal molecules. However, the driving voltages of Samples D3 and D4 have been increased compared to that of Sample D2, which may be due to the elevated content of rigid monomer (Bis-EMA15). Since the rigid monomer (Bis-EMA15) has a similar structure to the LC molecule, the increase in the rigid monomer (Bis-EMA15) content increases the anchoring force of the polymer matrix to the LC molecules, thus increasing the driving voltage. In contrast, the contrast of the film tends to increase first and then decrease as shown in Figure 9c. This is attributed to the change in the refractive index difference between the polymer matrix and the liquid-crystal molecule, resulting in a change in *T_on_*. In terms of response time, there is no significant difference between these samples. Overall, Sample D3 has good electro-optical performance and maintains good stability under long-term high-temperature conditions.

### 2.5. Electro-Optical Performance Test of PDLC Films under Different Temperature Conditions

Finally, the electro-optical properties of the PDLC film were tested and characterized under different temperature conditions, as shown in Figure 10a. From −20 °C to 80 °C, the PDLC film can be driven by a voltage to complete the scattering state to transparent state conversion. As we can see in Figure 10b, as the test temperature increases, the driving voltage of PDLC shows a significant decrease. This is due to the viscosity of LC decreasing with increasing temperature, which makes it easier to reorient the LC molecules in the electric field. Figure 10c shows the transition from the scattered to transparent state of the PDLC film driven by the electric field.

## 3. Experimental

### 3.1. Materials

The monomers hydroxypropyl methacrylate (HPMA), lauryl methacrylate (LMA), and polyethylene glycol diacrylate (PEGDA) were purchased from TCI (Shanghai) Development Co., Ltd., Shanghai, China. Bisphenol A ethoxylate dimethacrylate (Bis-EMA15) was purchased from Sigma Aldrich (Shanghai) Trading Co., Ltd., Shanghai, China. The photo-initiator Irgacure 651 was purchased from Beijing Kaiguo Technology Co. Ltd., Beijing, China. The nematic LC: SLC-1717, GXP-6011, and GXP-6015 were purchased from Yantai Xian Hua ChemTech. Co., Ltd., Yantai, China. The above materials were used without further purification. The chemical structures of monomers and photo-initiators, as well as the physical parameters of the liquid crystals, are shown in Figure 1. Glass spacers with a diameter of 20.0 (±1.0) μm were obtained from Sekisui Chemical Co., Ltd. (Osaka, Japan) to control the thickness of the PDLC films.

### 3.2. Preparation of the PDLC Films

First, the LC, polymerizable monomers, photo-initiator, and glass spacers are mixed in the proportions shown in Table 1. Subsequently, the milky white mixed solution is heated while stirring at high speed to mix uniformly until it appears isotropic and transparent. Then, the transparent mixed solution is sandwiched between two transparent conductive polyester substrates via capillary action. Finally, the samples are cured under UV light (365 nm, 10.0 mW/cm^2^) at room temperature (25 °C) for 10 min. After the preparation, PDLC films exhibit a strong light scattering state because of the disordered orientation of the LC molecules dispersed in the polymer matrix.

### 3.3. Measurements

*Morphological analysis*: the micro-morphology of the polymer network was observed via scanning electron microscopy (SEM, HITACHI S-4800, Tokyo, Japan). All samples were immersed into cyclohexane for about 7 days at room temperature to extract the LC molecules in the samples. Then, the samples were dried in a vacuum at 80 °C for 5 h. Finally, a thin layer of gold was coated via vapor deposition to increase conductivity.

*Shear-strength analysis*: the shear strength characterization of the PDLC films was tested by the universal tensile testing machine (LETRY) at the rate of 5 mm min^−1^. The area size of all tested PDLC films was 2.0 cm × 2.0 cm. The thickness of all tested PDLC films was 20 µm, which was controlled by the glass spacer.

*Electro-optical properties analysis*: the electro-optical properties of all samples were measured by a liquid-crystal device parameters tester (LCT-5016C, Changchun Liancheng Instrument Co. Ltd., Changchun, China) using a halogen laser beam (560 nm) as an incident light source. The transmittances of the samples were recorded by a photodiode, and the response of the photodiode was monitored by a digital storage oscilloscope. In the measurement, a square-wave-modulated electric field (100 Hz) was applied to cross the film. The transmittance of a blank LC cell was normalized as 100%.

*Analysis of working temperature*: the temperature control system was mounted on the LC device parameter test system to study the electro-optical properties of the PDLC films at different temperatures. The electro-optical properties of the PDLC films were tested from −20 °C to 80 °C in steps of 20 °C.

## 4. Conclusions

In this work, we systematically investigated the effects of LC content, different LC, chain length of crosslinker, and rigid monomer content on the microstructure, electro-optical properties, and thermal stability of PDLC films. Results revealed that the electro-optical properties of PDLC films can be optimized by the selection of suitable liquid-crystal species and crosslinkers with different chain lengths. On the other hand, the edge shrinkage problem of PDLC films was effectively improved when the content of rigid monomers was increased, which was attributed to the introduction of rigid monomers to enhance the strength of the whole polymer network. This work provides an effective strategy for improving the stability of PDLC films and optimizing their electro-optical properties, which is conducive to the development and large-scale application of PDLC films.

## Data Availability

Not applicable.

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
