# Peer review of "A Stable PDLC Film with High Ageing Resistance from an Optimized System Containing Rigid Monomer"

_molecules, 2023, doi:10.3390/molecules28041887_

Round 1

Reviewer 1 Report

Dear authors, you did excellent work with your paper. Your article offers very insightful information on developing PDLC films which can maintain stability at 80°C for 2000 hours.

The topic you choose in your paper is very interesting and challenging to discuss, which was very well-argued by all the polymer content, liquid crystals and monomer ratios that were optimized to obtain better electro-optical properties for the PDLC films.

This excellent paper provided relevant information regarding the film preparation, along with the composition of all samples. Also, the results provided relevant explanations based on the experiments you employed.

The paper's main aim was very well explained, describing the relevant film preparation and practical applications of these films in the field of smart windows.  The article is well-organized, structured and interesting. It also presents the successful development of a stable PDLC film with high mechanical strength and structural stability that can maintain its stability at 80°C for 2000 hours. The authors explained very thoroughly that by choosing liquid crystals with a wide temperature range, adopting acrylate polymer monomers with hydroxyl groups, and adjusting the polymer content, the formed PDLC film could function well from -20°C to 80°. Therefore, the main aim of the paper is being achieved. Therefore, by achieving these properties, the obtained films could have practical use in designing smart windows. Because there are relatively few studies in the field, the results presented by the authors, in fact, the effects of the introduction of rigid monomers within the film's composition, have significant outcomes regarding the mechanical and electro-optical properties, also the thermodynamics of the PDLC films investigated. Even if the paper is well structured, the introduction could be improved by adding other examples with relevant research in the field. Also, please consider improving the following: 1) Scheme 1- please, make it more visible, and describe what represents the first column (nematic LC?); 2) In Table 1, please delimitate each group with a line (after A5 from the group (A) by drawing a line). 3) The paragraph from 3.3. should be in the same form as the whole manuscript (justify); 4) Conclusion section could be enriched with relevant info related to results or other enhanced aspects. In my humble opinion, this article provides sufficient data to accept its current form. 

After reading and comprehending the information presented in the paper, I highly recommend it for publication.  

Keep up the excellent work!

Author Response

Response to Reviewer 1 Comments

Point 1: Scheme 1- please, make it more visible, and describe what represents the first column (nematic LC?).

Response 1: Thanks much for your comment. We have revised and supplemented Scheme 1 and replaced it in the revised version of the article.

Point 2: In Table 1, please delimitate each group with a line (after A5 from the group (A) by drawing a line.

Response 2: Thanks much for your comment. The Table 1 has been updated in the revised version of the article.

Point 3: The paragraph from 3.3. should be in the same form as the whole manuscript (justify).

Response 3: Thanks much for your comment. The paragraph formatting has been adjusted.

Point 4: Conclusion section could be enriched with relevant info related to results or other enhanced aspects.

Response 4: Thanks much for your comment. The conclusion section has been rewritten.

Reviewer 2 Report

The paper presents an extensive study of PDLC films’ properties vs quite large number of factors. The authors successively consider and optimize the LC content, LC type, monomer’s structure and rigidity regarding electro-optical performance, mechanical properties, and stability. The paper is well structured, the results are clearly presented, and also of practical use. I recommend to publish the paper, provided that the authors address the following remarks:

·      How was the films thickness controlled? As it is of critical importance, particularly for mechanical strength and the voltage, what was the thickness value and variation within a series of samples?

·      Can the authors comment on the voltage increase for C4-C5 comparing to C3 (Fig.6a,b) and for D3-D4 comparing to D2 (Fig.9a,b)?

·      PEGDA and Bis-EMA seem to be confused in a few places in a paragraph anticipating Fig.8. It can be seen from the figure that the mesh size increases with increase of Bis-EMA15 content (from D1 to D4), not vice versa. Also, the molecular weight of PEGDA700 is not higher than that of Bis-EMA15, which makes sense as the mesh increases with Bis-EMA15 content. The trend is valuable as it demonstrates that the monomer rigidity beats the mesh size effect in terms of film durability.

Minor corrections:

·      Scheme 1 caption should add “Chemical structures and properties of the materials used …”

·      In a structure of Bis-EMA (scheme 1) an Oxygen is missing in the left acrylate group

·      References should be added for Eqs 1 and 2. When the corresponding parameters are denoted (p. 5), Deis missing. R should be denoted as well as it is intensively referred to afterwards

·      Minor typo correction is needed

Author Response

Response to Reviewer 2 Comments

Point 1: How was the films thickness controlled? As it is of critical importance, particularly for mechanical strength and the voltage, what was the thickness value and variation within a series of samples?

 Response 1: Thanks much for your comment. The thickness of the films was controlled by the glass spacer of 20 µm. In the preparation of the PDLC films, glass spacers are mixed with monomers and liquid crystals and then sandwiched between two layers of ITO conductive films. Relevant information and preparation methods have been updated in the revised article.

The difference in thickness can have a large impact on the mechanical and electro-optical properties of the PDCL films. In practical applications, especially in the field of architectural windows, 20 µm thickness is a more general standard, so this paper chooses 20 µm as the unified.

Point 2: Can the authors comment on the voltage increase for C4-C5 comparing to C3 (Fig.6a,b) and for D3-D4 comparing to D2 (Fig.9a,b)?

Response 2: Thanks much for your comment. We can see the SEM photos in Fig.5, the mesh size in C4 and C5 are larger than C1-3. Generally, the driving voltage is negatively correlated with the mesh size due to the reduced anchoring force of the polymer matrix on the LC molecules. Based on this, we infer that the increase in driving voltage of C4 and C5 may receive the influence of the chain length of the crosslinker molecules. As the chain length of cross-linker molecules increases, their spatial site resistance increases and their reactivity decreases, which increases the probability that the cross-linker molecules remain in the liquid crystal microdroplets, thus affecting the electro-optical properties of the liquid crystal molecules.

The voltage increase for D3-D4 may be due to the elevated rigid monomer (Bis-EMA15) content. Since the rigid monomer has a similar structure to the LC molecule, the increase in the rigid monomer content increases the anchoring force of the polymer matrix to the LC molecules, thus increasing the driving voltage.

The corresponding content has been updated in the revised version of the article

Point 3: PEGDA and Bis-EMA seem to be confused in a few places in a paragraph anticipating Fig.8. It can be seen from the figure that the mesh size increases with increase of Bis-EMA15 content (from D1 to D4), not vice versa. Also, the molecular weight of PEGDA700 is not higher than that of Bis-EMA15, which makes sense as the mesh increases with Bis-EMA15 content. The trend is valuable as it demonstrates that the monomer rigidity beats the mesh size effect in terms of film durability.

Response 3: Thanks much for your comment. In general, the smaller the size of the polymer matrix mesh, the better its mechanical properties should be. In D3 to D4, the polymer mesh size becomes larger, but its stability increases, which proves that the rigid monomer improves the stability more than the mesh size effect in this system. The discussion of this section has been added to the article.

Point 4: Scheme 1 caption should add “Chemical structures and properties of the materials used …”

  • In a structure of Bis-EMA (scheme 1) an Oxygen is missing in the left acrylate group
  • References should be added for Eqs 1 and 2. When the corresponding parameters are denoted (p. 5), Deis missing. R should be denoted as well as it is intensively referred to afterwards
  • Minor typo correction is needed

Response 4: Thanks much for your comment. The relevant issues mentioned above have been revised and updated in the revised article.

Reviewer 3 Report

In this work, a PDLC film with high mechanical strength and structural stability was prepared, which can maintain stability at 80 °C for 2000 hours. By choosing liquid crystals with a wide temperature range, adopting acrylate polymer monomers containing hydroxyl groups, and adjusting the polymer content, the PDLC film can work well from -20 °C to 80 °C. On this basis, the effects of the introduction of rigid monomers on the mechanical properties and electro-optical properties of PDLC films were investigated. Authors have systematically carried out the experiments and presented the well-organized data. The work is interesting and can be considered for publication in Molecules after addressing minor comments.

1.     I recommend the authors to include some new references of Molecules which are relevant to use in introduction as well as results and discussion.

2.     Recheck and reformat the figures to match the standards of the Molecules.

3.     There are so many typographic errors throughout the manuscript as well as in figures. Authors must revise the manuscript and make it error free.

4.     How is the stability of these Materials?

5.     The conclusion should be brief and informative. Rewrite the conclusion.

Author Response

Response to Reviewer 3 Comments

Point 1: I recommend the authors to include some new references of Molecules which are relevant to use in introduction as well as results and discussion.

 Response 1: Thanks much for your comment. We have updated 5 related references as follows.

  1. Chen, H.; Liu, Y.; Chen, M.; Jiang, T.; Zhang, L.; Yang, Z.; Yang, H. Research of Liquid-Crystal Materials for a High-Performance FFS-TFT Display. Molecules 2023, 28, doi:10.3390/molecules28020754.
  2. Wu, Q.; Zhang, H.; Jia, D.; Liu, T. Recent Development of Tunable Optical Devices Based on Liquid. Molecules 2022, 27, doi:10.3390/molecules27228025.
  3. Zhao, Y.; Li, J.; Yu, Y.; Zhao, Y.; Guo, Z.; Yao, R.; Gao, J.; Zhang, Y.; Wang, D. Electro-Optical Characteristics of Polymer Dispersed Liquid Crystal Doped with MgO Nanoparticles. Molecules 2022, 27, doi:10.3390/molecules27217265.
  4. Zhang, W.; Nan, Y.; Wu, Z.; Shen, Y.; Luo, D. Photothermal-Driven Liquid Crystal Elastomers: Materials, Alignment and Applications. Molecules 2022, 27, doi:10.3390/molecules27144330.
  5. Feng, Y.-Q.; Lv, M.-L.; Yang, M.; Ma, W.-X.; Zhang, G.; Yu, Y.-Z.; Wu, Y.-Q.; Li, H.-B.; Liu, D.-Z.; Yang, Y.-S. Application of New Energy Thermochromic Composite Thermosensitive Materials of Smart Windows in Recent Years. Molecules 2022, 27, doi:10.3390/molecules27051638.

Point 2: Recheck and reformat the figures to match the standards of the Molecules.

Response 2: Thanks much for your comment. The formatting of the images and tables in the article has been checked.

Point 3: There are so many typographic errors throughout the manuscript as well as in figures. Authors must revise the manuscript and make it error free.

Response 3: Thanks much for your comment. The the manuscript has been checked and revised.

Point 4: How is the stability of these Materials?

Response 4: Thanks much for your comment. The acrylate materials are very stable, and films prepared from the same system can have a service life of several decades

Point 5: The conclusion should be brief and informative. Rewrite the conclusion.

Response 5: Thanks much for your comment. The conclusion section has been rewritten.
